


# 1 Quantifying the soil sink of atmospheric Hydrogen: a full year of field
# 2 measurements from grassland and forest soils in the UK

Nicholas Cowan[1], Toby Roberts[1], Mark Hanlon[1], Aurelia Bezanger[1], Galina Toteva[1,2], Alex Tweedie[1,2], Karen
Yeung[1], Ajinkya Deshpande[1], Peter Levy[1], Ute Skiba [1], Eiko Nemitz[1], Julia Drewer[1]
[1]UK Centre for Ecology and Hydrology, Easter Bush, Midlothian, UK, EH26 0QB
[2]School of GeoSciences, The University of Edinburgh, Edinburgh, United Kingdom
**Corresponding author:** Nicholas Cowan (nicwan11@ceh.ac.uk)
**Keywords:** greenhouse gas, carbon, methane, flux, chamber methodology

## 9 Abstract

Emissions of hydrogen ($H_2$) gas from human activities are associated with indirect climate warming effects. As
the hydrogen economy expands globally (e.g. the use of $H_2$ gas as an energy source), the anthropogenic
release of $H_2$ into the atmosphere is expected to rise rapidly as a result of increased leakage. The dominant
$H_2$ removal process is uptake into soils; however, removal mechanisms are poorly understood and the fate
and impact of increased $H_2$ emissions remains highly uncertain. Fluxes of $H_2$ with soils are rarely measured,
and data to inform global models is based on few studies. This study presents soil $H_2$ fluxes from two field
sites in central Scotland, a managed grassland and a planted deciduous woodland, with flux measurements
of $H_2$ covering full seasonal cycles. A bespoke flux chamber measurement protocol was developed to deal
with the fast decline in headspace concentrations associated with rapid $H_2$ fluxes, in which non-linear
regression models could be fitted to concentration data over a 7-minute enclosure time. We estimate annual
$H_2$ uptake of -3.1 ± 0.1 and -12.0 ± 0.4 kg $H_2$ ha$^{-1}$ yr$^{-1}$ and mean deposition velocities of 0.012 ± 0.002 and
0.088 ± 0.005 cm s$^{-1}$ for the grassland and woodland sites, respectively. Soil moisture was found to be the
primary driver of $H_2$ uptake at the grassland site, where the high clay content of the soil resulted in anaerobic
conditions (near zero $H_2$ flux) during wet periods of the year. Uptake of $H_2$ at the forest site was highly variable
and did not correlate well with any localised soil properties (soil moisture, temperature, total carbon and
nitrogen content). It is likely that the high clay content of the grassland site (55% clay) decreased aeration
when soils were wet, resulting in poor aeration and low $H_2$ uptake. The well-drained forest site (25% clay) was
not as restricted by exchange of $H_2$ between the atmosphere and the soil, showing instead a large variability
in $H_2$ flux that is more likely to be related to heterogeneous factors in the soil that control microbial activity
(e.g. labile carbon and microbial densities). The results of this study highlight that there is still much that we



do not understand regarding the drivers of $H_2$ uptake in soils and that further field measurements are required
to improve global models.

## 1. Introduction

Prior to the industrial revolution in the 18th century, the atmospheric concentration of Hydrogen gas ($H_2$) was
relatively stable at approximately 330 ppb (Patterson et al., 2021). Human activity over the past two centuries
has resulted in increasing atmospheric $H_2$ concentrations (546 ppb in 2021, Petron et al. (2023)), partly as a
result of increasing industrial leaks (Hitchcock 2019; Cooper et al., 2022), partly due to increases in emissions
and concentrations of precursor gases such as methane ($CH_4$) and volatile organic compounds (VOCs), and
partly due to increasing concentrations of other gases in the atmosphere which extend the natural lifetime
of $H_2$ (Patterson et al., 2021).  In the atmosphere, $H_2$ competes for hydroxyl (OH) radicals with gases such as
methane ($CH_4$) and carbon monoxide (CO), thus an increase in concentrations of these gases due to human
activities has resulted in increasing competition for OH and extended the lifetimes for each species (Khalil &
Rasmussen, 1990; Bertagni et al., 2022). Concentrations of atmospheric $H_2$ gas are indirectly associated with
climate warming effects as a result of extending the atmospheric lifetime of the powerful greenhouse gas $CH_4$
as well as increasing tropospheric ozone and water vapour, which also have a warming potential (Warwick et
al., 2004; Ocko & Hamburg, 2022). The associated indirect global warming potential (GWP) had been
estimated to be in the range of 3.3 to 5 over a hundred-year time horizon (Derwent et al., 2020, Field &
Derwent, 2021), though recent estimates have been made of up to 11.6 ± 2.8 times that of an equivalent
mass of carbon dioxide (Sand et al., 2023). The effective GWP and the atmospheric accumulation of $H_2$ are
highly sensitive to its atmospheric lifetime, which is estimated to be approximately 2 years (Novelli et al.,

50  1999).

The dominant process for $H_2$ removal from the atmosphere is uptake by soils, which is estimated to be three
times larger than the sink due to atmospheric reaction with OH (Warwick et al., 2004; Derwent et al., 2020;
Field & Derwent, 2021; Paulot et al., 2021; Ocko & Hamburg, 2022). Whilst both removal mechanisms are
highly uncertain, the fate and impact of increased $H_2$ emissions depends largely on the soil sink strength
(Ehhalt & Rohrer, 2009). The soil $H_2$ sink is caused by microbial activity, both under aerobic and anaerobic
conditions (Piché-Choquette & Constant, 2019). A large spectrum of bacteria and archaea can utilise $H_2$ as an
energy source, via the hydrogenase enzyme. Whilst some investigations have highlighted the importance of
high-affinity $H_2$-oxidising bacteria (Saavedra-Lavoie et al., 2020), most studies suggest that this enzyme is
widespread across many bacterial and archaeal phyla, and that $H_2$ consumption is the norm rather than the
exception (Islam et al., 2020; Greening & Grinter, 2022). It has been suggested that the potential soil $H_2$ sink
is very large because of the high $H_2$ demand of microbes (Smith-Downey et al., 2008). However, specific $H_2$





uptake rates for different soil types and conditions are lacking. In addition to microbial activity, diffusion into
the soil is a further important rate limiting step. Gases penetrate the soil by passive diffusion and diffusion
rates are mainly influenced by porosity, which is affected by soil structure, texture, organic matter contents,
vegetation types (roots) and moisture content. Thus, for the same microbial activity, porous soils can be
expected to be much larger $H_2$ sinks than compacted and/or waterlogged soils due to increased gas exchange
rates with the atmosphere. At the larger scale, diffusion rates will depend on the changing climate: a wetter
climate may lower the $H_2$ diffusion rates (Paulot et al., 2021). Temperature is another important factor as it
determines the rate of microbial enzyme reactions, and a carbon source is required for heterotrophic
microbial activity (Islam et al., 2020; Meredith et al., 2016; Baril et al., 2022). In addition, soil $H_2$
concentrations will be competing with $CH_4$ as the energy source for soil microbes, hence the $H_2$ sink strength
may in turn affect the $CH_4$ sink strength and vice versa (Conrad, 1999). The biological sink of atmospheric $H_2$
has been suggested to be more sensitive to spatial variations of drivers compared to the fluxes of other gases
with high variability such as nitrous oxide ($N_2O$); however, $H_2$ measurement data are limited (Baril et al.,

75    2022).

Historically, the processes that control $H_2$ uptake in soils have been severely understudied due to the logistical
difficulties and technical constraints on measuring $H_2$ fluxes. This study presents measurements of $H_2$ fluxes
between the soil and the atmosphere at two field sites in central Scotland, a managed grassland and a planted
deciduous woodland. These are the first reported flux measurements of $H_2$ covering a full annual cycle in the
UK. It has previously been reported that forest ecosystems exhibit higher $H_2$ uptake rates than agroecosystems
(Ehhalt and Rohrer, 2009); however, the generality of this and exact mechanisms are still unclear. This study
aims to investigate the response of microbial $H_2$ uptake at a grassland and a forest site to environmental
drivers, and to identify differences between the sites. We also describe a dedicated flux chamber
methodology which has been developed to best address the challenges of measuring $H_2$ flux using gas
chromatography (GC) analysers.

## 2. Methods

### 2.1.   Field Sites

Measurements of trace gas fluxes and environmental variables were made at two field sites within the
Midlothian region in central Scotland (UK, approximately 6 miles south of Edinburgh, Table 1). The first of
these was the long-term environmental monitoring site at Easter Bush Farm (grassland). The grassland site
(55.8653 °N, -3.206 °W) is an intensively managed, improved grassland (South field in Cowan et al., 2020 and
Drewer et al., 2016) that since 2001 has been used predominantly to graze sheep, with a species composition



of >99% perennial ryegrass (*Lolium perenne*). The soil type is an imperfectly drained Eutric Cambisol with clay
soil. The field management is typical for this region, with predominately ammonium nitrate (AN) fertilisation
via tractor-mounted broadcast spreading, with liming every 3 – 5 years to maintain the pH between 5.5 and
6.0 and occasional ploughing and reseeding. The sheep were absent from the fields in the winter months
(November to February), with sporadic movement between local fields throughout the growing season
(March to September) as management required. During the period of 01/10/23 to 01/10/24, the cumulative
rainfall at the grassland site was 1133 mm and the mean temperature was 8.6 °C which is fairly typical of the
site (Table 1)
The second field site was a temporary experimental area setup in Glencorse Forest (woodland). Glencorse
Forest (55.8540°N, -3.215°W) was converted to a planted deciduous forest from a pasture approximately 40
years prior to measurements (Billington and Pelham, 1991). The study plot is situated in a plantation of Silver
Birch (*Betula pendula*) and Downy Birch (*Betula pubescens*), with a ground flora consisting mostly of grasses.
The soil is classified as a sandy loam which lies under a thin layer (5 – 10 mm) of organic debris. The field site
had been subject to enhanced nitrogen deposition with ammonia for approximately 2 years before $H_2$ flux
measurements were carried out (Deshpande et al., 2024). During the period of 01/10/23 to 01/10/24, the
cumulative rainfall at the woodland site was 1047 mm and the mean temperature was 9.6 °C which was
slightly wetter and warmer than historical mean data (Table 1).
**Table 1** Field site environmental properties as reported in previous studies and ongoing research. Mean
annual values taken from 10+ years of site data. Rainfall represents throughfall (e.g. rain that reaches the
soil).

| Property | Easter Bush Farm | Glencorse Forest |
|---|---|---|
| Management | Improved grassland (grazed) | Planted woodland (Birch) |
| Abbreviation | Grassland | Woodland |
| Soil Type | Mineral | Mineral |
| Carbon Content (% mass) | 4.0 | 3.1 |
| pH | 5.5 | 5.3 |
| Bulk Density (g cm$^{-3}$) | 1.11 | 0.96 |
| Particle Density (g cm$^{-3}$) | 2.57 | 2.34 |
| Sand/silt/clay (%) | 25/20/55 | 60/15/25 |
| Mean Annual Temperature (°C) | 8.4 | 9.0 |
| Mean Annual Rainfall (mm) | 1040 | 920 |


### 2.2.    *Meteorological and soil measurements*

Continuous environmental measurements were made at both field sites. Air temperature, soil temperature,
soil volumetric water content (VWC) at three depths (5, 10 and 20 cm at the grassland site; 5, 10 and 15 cm



at the woodland site), relative humidity (RH) and rainfall were measured at both sites throughout the flux
measurement campaign (Table S1). For each flux chamber measurement, soil temperature and soil VWC were
also measured next to the chamber (<0.5 m distance) at the time of the flux measurement. Soil temperature
was measured at 10 cm depth using a handheld probe (ETI Ltd., Worthing, UK), and soil VWC was measured
at 12 cm depth using an HS2 HydroSense II handheld soil moisture sensor (Campbell Scientific, Utah, USA),
with 4 replicates for each chamber. Soil samples were collected for total carbon (C) and total nitrogen (N)
analysis from the top 10 cm of soil at the woodland site in March 2021, September 2021, May 2022, August
2022, November 2022, and March 2023. Subsamples were dried at 105 °C until constant weight, milled using
a ball mill (MM200 ball mill, Retsch, Haan, Germany) and analysed using an elemental analyser (Flash SMART,
Thermo Fisher Scientific, MA, USA).

### 2.3.    Flux measurements

Fluxes of hydrogen ($H_2$), methane ($CH_4$) and nitrous oxide ($N_2O$) were measured using the static chamber
method (e.g. Drewer et al., 2016). Chambers (diameter = 40 cm, height = 30 cm) consisting of opaque
polypropylene open-ended cylinders, were installed at each field site: 20 at Easter Bush (grassland) and 36 at
Glencorse (woodland). The chambers were inserted into the ground to a depth of approximately 10 cm for
the entire study period. The depth to the surface in each chamber was measured at 5 points on the sides of
the chamber base using a ruler, from which the average was used to calculate the volume of air within. During
measurement periods, aluminium lids were fastened onto the bases using four strong clips; a strip of draft
excluder glued onto the lid provided a gas tight seal between chamber and lid.  A three-way tap was used for
gas sample removal using a 100 ml syringe. 20 ml glass vials were filled with a double needle system to flush
the vials with five times their volume. Storage tests using gas standards revealed that gases stored in the vials
were stable for up to 24 hours, after which $H_2$ leakage could be observed in the data. Hence all analyses of $H_2$
gas samples from the chambers were carried out within 24 hours of measurement in the field (typically within
6 hours). Measurements of $H_2$ and GHGs were made approximately monthly.
Two separate measurement protocols were employed to measure greenhouse gases (GHGs) and $H_2$ fluxes,
due to the differences in how the gases behaved within the chamber over a given timespan. For GHG
measurements, the standard practice of extracting four gas samples (100 ml) at regular intervals over one
hour (0, 20, 40, 60 min) was used (Drewer et al. 2017). However, due to the rapid uptake of $H_2$ observed in
trial measurements ($H_2$ in the chamber headspace could reach zero ppb in under 10 mins), the time-evolution
of $H_2$ in the chamber was non-linear and therefore a separate measurement protocol was developed for $H_2$
fluxes. Fluxes of $H_2$ were measured during entirely separate enclosure periods to the GHGs (albeit on the
same day) using an enclosure period with 6 samples taken over 7 minutes (0, 1, 2, 3, 5 & 7 mins). Chambers



used to measure $H_2$ were fitted with a small 5 cm diameter PC fan which ran from a 9 V battery during chamber
enclosure times to ensure rapid air mixing over the shorter measurement period.
Concentrations of $H_2$ were measured using an Agilent 8890 gas chromatograph fitted with a pulsed discharge
helium ionization detector (GC-PDHID) equipped with a 7697A headspace autosampler, with capacity for 108
vials (Agilent, Santa Clara, California, USA). Concentrations of $CH_4$ and $N_2O$ were measured using a gas
chromatograph (Agilent 7890B with headspace autosampler 7697A with capacity for 108 vials; Agilent, Santa
Clara, California, USA) with a micro-electron capture detector (μECD) for $N_2O$ analysis and flame ionization
detector (FID) for $CH_4$ analysis run in parallel. Each analytical run of $H_2$ and GHG samples included at least
three sets of four certified standard concentrations for calibration purposes (certified to ± 5%). The
instrumental noise (σ) of the instruments were 40, 5, and 15 ppb for $CH_4$, $N_2O$ and $H_2$, respectively. Based on
the methods used, the analytical uncertainty in flux estimates were 0.38, 0.047 and 1.08 nmol m$^{-2}$ sec$^{-1}$ for
$CH_4$, $N_2O$ and $H_2$, respectively.
Fluxes were calculated using linear and non-linear regression methods using the HMR package for the
statistical software R (Pedersen *et al.*, 2010). By convention, positive fluxes represent emission from the soil,
and negative fluxes indicate that the soil acts as a sink. Fluxes of GHGs were all calculated using linear
regression, where *dC/dt* is calculated using the standard line of best fit through the concentration data. As
concentrations of $H_2$ fall exponentially during chamber measurements when soil uptake of $H_2$ is high, linear
regression is not always appropriate. To account for this, fluxes of $H_2$ were calculated using both linear
regression and the HM model, depending on the magnitude of the rate of change observed in each chamber
measurement. The HM model is a commonly used non-linear model derived by Hutchinson & Mosier (1981)
with a negative exponential form of curvature which calculates the rate of change of a gas concentration at
*t* = 0. The concentration *C* at time *t* is given by Equation 1, where $C_0$ is the initial concentration, $C_{max}$ is the
value at equilibrium and *k* is a constant. *dC/dt* is is the initial rate of change in concentration at *t* = 0 in nmol
mol$^{-1}$ s$^{-1}$, calculated using Equation 2.
$$C_t = C_{max} - (C_{max} - C_o)\exp(-kt) \qquad \text{(Equation 1)}$$

$$\frac{dC}{dt} = k(C_{max} - C_o) \qquad \text{(Equation 2)}$$

The initial *dC/dt* is used to calculate the flux using Equation 3, where *F* is gas flux from the soil (nmol m$^{-2}$ s$^{-1}$),
ρ is the density of air in mol m$^{-3}$, *V* is the volume of the chamber in m$^3$ and *A* is the ground area enclosed by
the chamber in m$^2$.
$$F = \frac{dC}{dt} \times \rho \times \frac{V}{A} \qquad \text{(Equation 3)}$$



At low concentrations near the limit of detection of the analyser, a clear exponential decline was hard to
discern from the measurement noise and could give rise to spurious fits to Equation 1. (Examples 1 and 2 in
Figure 1 and Table 2). The criteria for using the HM model for each individual flux calculation was based on i)
$k$ is not unrealistically large in Equation 2 (as defined and limited by the HMR package in R), ii) the flux
estimated by linear regression is larger than the analytical uncertainty of the method (1.08 nmol m$^{-2}$ s$^{-1}$ for
H$_2$) and iii) the 95 % confidence interval (95% C.I.) of the HM model fit is less than 5 times the magnitude of
the flux estimated using linear regression (removes poor-fitting outliers). In Figure 1 and Table 3, six examples
are given in which three selections of linear regression fitting and three selections of the HMR model fitting
are used to determine flux. For large uptake fluxes (Examples 4, 5 and 6) the HMR model provides a more
suitable fit to the non-linearity in $dC/dt$, which linear regression does not accurately represent. Deposition
velocity of H$_2$ was calculated by dividing the calculated flux by the ambient concentration at the site (mean
of $t$ = 0 measurements on day of measurement in mol m$^{-3}$).

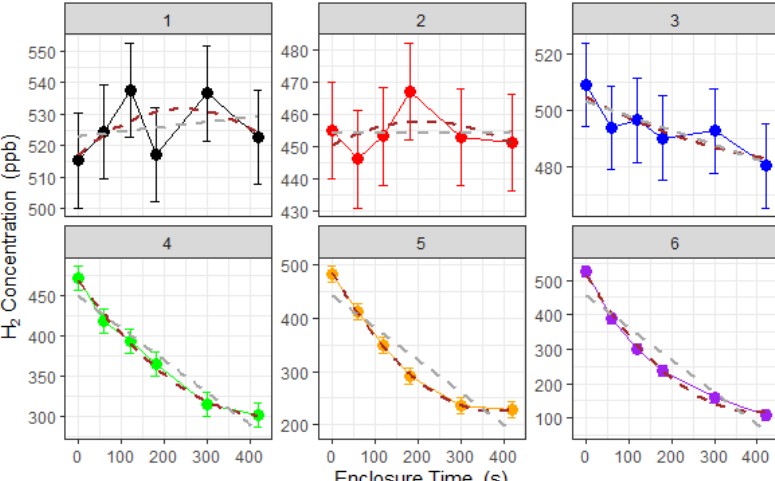


**Figure 1.** Examples of concentration data collected during H$_2$ flux chamber sampling. Linear regression (grey)
and HM model (brown) are used to determine $dC/dt$ for each chamber measurement. Error bars represent
the analytical uncertainty of H$_2$ measurements by GC analysis (15 ppb in this study). Comparisons of flux data
presented in Table 2.



**Table 2.** Further information on the example data provided in Figure 1. Six examples of chamber $H_2$ flux measurements are provided, from the Easter Bush (grassland) and Glencorse (woodland) field sites. The initial and final concentrations of $H_2$ within the chamber are provided, as well as the flux and 95% C.I. calculated using linear and HM model (Equation 2) fitting methods (NA when $k$ is too large). The method selected to represent the flux in this study based on the described protocols is included.

| Example | Date | Location | Initial (ppb) | Final (ppb) | Flux Linear fit (nmol m$^{-2}$ s$^{-1}$) | Flux HM fit (nmol m$^{-2}$ s$^{-1}$) | Selected Method |
|---------|------|----------|---------------|-------------|------------------------------------------|--------------------------------------|-----------------|
| 1 | 10/04/2024 | Grassland | 515 | 522 | 0.01 (-0.59 – 0.63) | 2.839 (NA) | Linear |
| 2 | 16/11/2023 | Grassland | 455 | 451 | 0.003 (-0.56 – 0.60) | 0.239 (-6.47 – 6.99) | Linear |
| 3 | 13/02/2024 | Grassland | 509 | 480 | -0.319 (-0.58 – -0.06) | -0.889 (-2.60 – 0.21) | Linear |
| 4 | 31/07/2024 | Grassland | 471 | 300 | -3.078 (-4.54 – -3.35) | -6.6 (-9.44 – -3.80) | HM |
| 5 | 31/07/2024 | Grassland | 483 | 229 | -3.152 (-4.54 – -3.35) | -10.89 (-15.54 – -6.232) | HM |
| 6 | 04/04/2024 | Woodland | 527 | 109 | -5.278 (-7.05 – -1.07) | -14.35 (-15.88 – -12.82) | HM |

## 3. Results

### 3.1. *Hydrogen Flux measurements*

Fluxes of $H_2$ measured from the grassland site ranged from -15.5 to +5.3 nmol m$^{-2}$ s$^{-1}$ (deposition velocity (Vd) ranged from 0.070 to -0.026 cm s$^{-1}$) (Figures 2 and S1) over the period of September 2023 to September 2024. More than 90% of the $H_2$ fluxes measured at the grassland site were negative (soil uptake) and only 2 of 251 chamber measurements showed emissions from the soil which exceed the analytical uncertainty of the method. Fluxes of $H_2$ at the grassland site changed seasonally, with greater uptake in the spring and summer compared with winter, where the flux was close to zero. Fluxes at the grassland site had a median of -1.2 nmol m$^{-2}$ s$^{-1}$ and 95% percentiles of -9.9 to 0.2 nmol m$^{-2}$ s$^{-1}$. Fluxes measured from the woodland site ranged from -40.7 to -1.1 nmol m$^{-2}$ s$^{-1}$ (Vd ranged from 0.191 to 0.005 cm s$^{-1}$) (Figures 2 and S1). All fluxes measured at the woodland site showed $H_2$ uptake in the soil. Spatial variability of $H_2$ flux at the woodland site was an order of magnitude larger than those observed at the grassland site. Fluxes at the woodland site had a median of -18.7 nmol m$^{-2}$ s$^{-1}$ and 95% percentiles of -32.4 to -4.3 nmol m$^{-2}$ s$^{-1}$. Ambient concentrations of $H_2$ at the sites ranged from 424.8 to 566.5 ppb. Mean ambient concentrations at the woodland site (484.4 ppb) were on average 21.7 ppb (4.3 %) lower than the grassland site (506.5 ppb) which could be considered statistically insignificant (t-test, $p > 0.1$), but differences were fairly consistent throughout the year (summary statistics presented in Table S2).





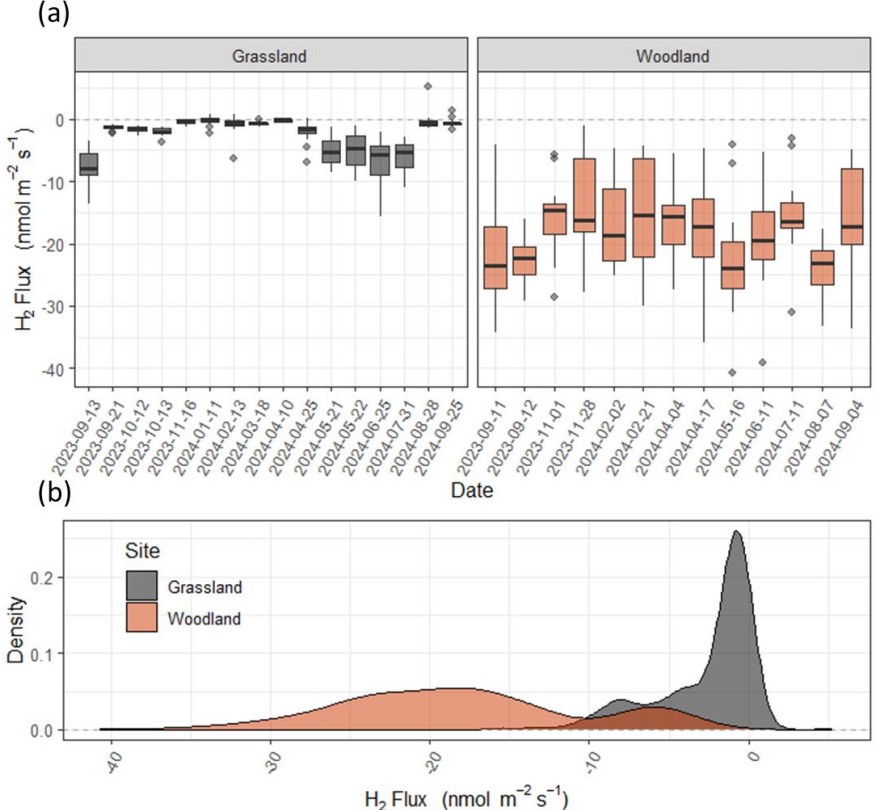

223

**Figure 2.** Fluxes of $H_2$ measured using the flux chamber method at grassland (Easter Bush, grassland; grey)

and forest (Glencorse Forest, woodland; red) sites in Midlothian, Scotland. Boxplots (a) represent the median,

and 25[th] and 75[th] percentiles of flux data of 20 chambers, respectively (whiskers represent the 95[th]

percentiles). (b) Frequency distributions of the flux data for both sites (Figure replicated for Vd in Figure S1).

### 3.2.  *Greenhouse gas fluxes*

Fluxes of $CH_4$ at both sites were close to zero, with mostly small negative fluxes observed at both sites (Figure

S3). Soil uptake of $CH_4$ was observed during the summer months at both sites but during colder months, only

the woodland site continued to observe consistent negative $CH_4$ fluxes. Fluxes of $CH_4$ measured from the

grassland site ranged from -1.2 to 1.0 nmol m$^{-2}$ s$^{-1}$ with a median of -0.14 nmol m$^{-2}$ s$^{-1}$. Fluxes of $CH_4$ measured

from the woodland site ranged from -1.3 to 2.3 nmol m$^{-2}$ s$^{-1}$ with a median of -0.32 nmol m$^{-2}$ s$^{-1}$. Only 40% of

all $CH_4$ flux measurements exceeded the analytical uncertainty of the chamber method deployed, highlighting

the magnitude of observed fluxes were near the limit of detection of the methodology. Fluxes of $N_2O$

measured at both sites were relatively low for all measurement dates (58% of all data below the analytical





uncertainty) with the exception of measurements made in April at the grassland site. Nitrogen fertiliser was
applied to the field on the 28th of March, resulting in increased $N_2O$ emissions for several weeks (Figure S3).

### 3.3. Drivers of $H_2$ flux

Correlations of $H_2$ flux with soil moisture and soil temperature can be observed at both sites (Figures 4a &
4b); however, each site responds differently. Fluxes of $H_2$ at the grassland site were close to zero when water
filled pore space (WFPS) was high (>45%), then tended towards uptake as WFPS decreased. The correlation
between $H_2$ flux and WFPS is weaker at the woodland site and flux data are widely scattered. Fluxes of $H_2$ at
both the grassland and woodland site tended towards higher uptake as temperature increased, though
scatter increased toward higher uptake at both sites (>12 °C). A simplistic multiple regression fit between $H_2$
flux (y) with soil moisture (x) and soil temperature (z) ($y = a_1x^2 + a_2x + b_1z^2 + b_2z + c$) accounts for more than
half of the variance in the observed fluxes at the grassland site ($R^2 = 0.60$) with a significant contribution from
soil moisture, but the same approach does not adequately represent the large flux variability at the woodland
site ($R^2 = 0.14$) for which neither soil moisture or soil temperature was found to correlate significantly (Table
S3). Fluxes of $CH_4$ at the sites followed the same trends as $H_2$ flux in terms of emission/uptake and follow
similar correlations with soil moisture and soil temperature as $H_2$ flux (Figures 4c & 4d). Fluxes of $CH_4$ at both
sites were close to zero (or emission) when soils were wet (>45 % WFPS) and cold (<6 °C). Uptake of $CH_4$ was
greatest when soils were drier and warm.
Total carbon (C) and total nitrogen (N) from the woodland site provided comparisons of $H_2$ flux with soil C and
N at the chamber level (Figure S4). Variability in C and N in the replicated cores per chamber was similar to
the magnitude of spatial variability at the plot scale, suggesting that localised soil samples were not
adequately representative of the soil within a chamber (high spatial variability of C and N in the soil at the
<1 $m^2$ scale). No correlation between $H_2$ flux with measured total soil C or N in the top 10 cm was found at
the woodland site ($R^2 < 0.01$ for each).
By combining continuous soil measurement data collected at each site (soil moisture and temperature at 10
cm depth), with the multiple regression model with soil moisture and soil temperature (Figures 4b & 4c) as
described in Table S1, continuous $H_2$ flux predictions were made for a full year (Figure 4a).  This model predicts
that $H_2$ flux at the grassland site remains close to zero for most of the time, except when soil moisture drops
(e.g. warm months in spring and summer). The model predicts that $H_2$ flux at the grassland site is strongly
dependent on the soil moisture content, with relatively strong periods of $H_2$ uptake during drier periods
(warm periods between rainfall events). $H_2$ flux estimates at the woodland site are more variable, and less
susceptible to changes in meteorology or soil conditions. The model predicts a slowdown in $H_2$ uptake in the



forest soils during the colder months in winter but is not significantly impacted by changing soil moisture.
Total annual estimates of $H_2$ flux predicted by the model are -3.1 ± 0.1 and -12.0 ± 0.4 kg $H_2$ ha$^{-1}$ yr$^{-1}$ for the
grassland and woodland sites, respectively. By comparison, a straight average of the measurements, without
using models to gap-fill the data, suggests mean fluxes (with 95% C.I.s) of -2.6 ± 0.4 and -18.7 ± 1.0 nmol m$^{-2}$
s$^{-1}$ which would translate to annual cumulative fluxes of approximately -1.6 ± 0.2 and -11.7 ± 0.6 kg $H_2$ ha$^{-1}$ yr$^{-}$
$^1$ for the grassland and GC sites, respectively. The two estimates agree well at the woodland site, but the gap
filling increases the estimated annual $H_2$ uptake at the grassland site by 56%.

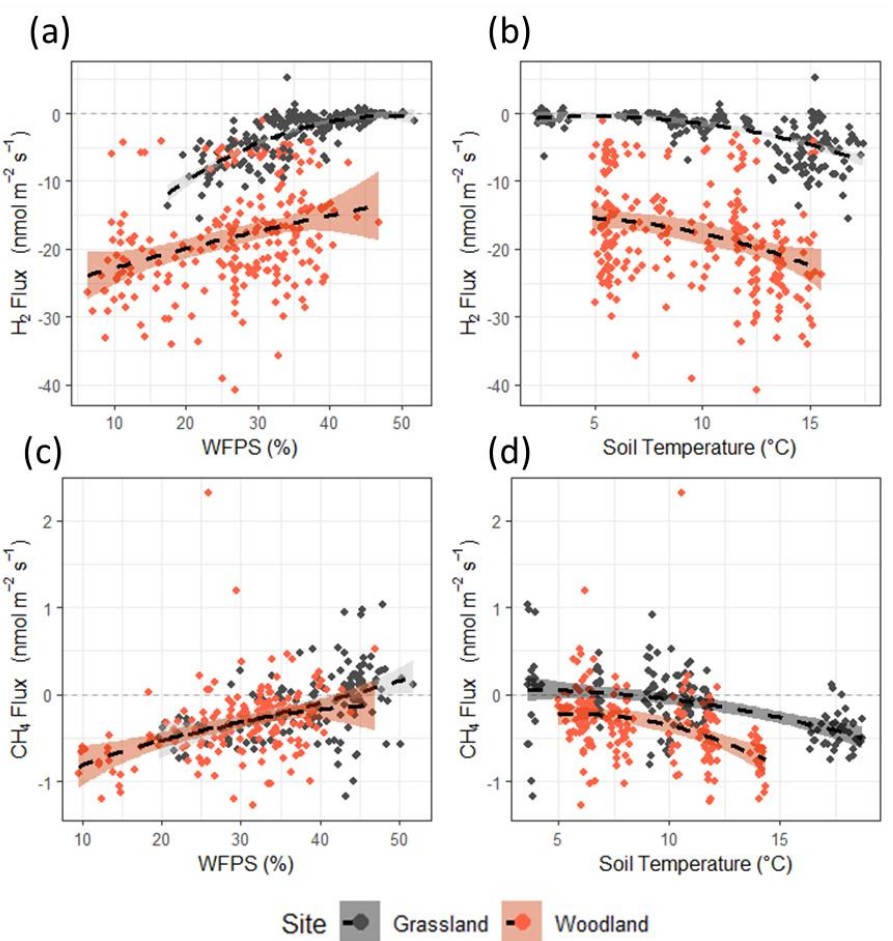

**Figure 3.** Correlations between $H_2$ flux and (a) water filled pore space (WFPS) and (b) Soil Temperature.
Correlations between $CH_4$ flux and (c) water filled pore space (WFPS) and (d) Soil Temperature. WFPS and soil



temperature measured at 10 cm depth via sampling probe. A 2nd order polynomial fit (black dashed line) is
included as a visual aid ($y = a_1x^2 + a_2x + c$) (Figure replicated for Vd in Figure S2).

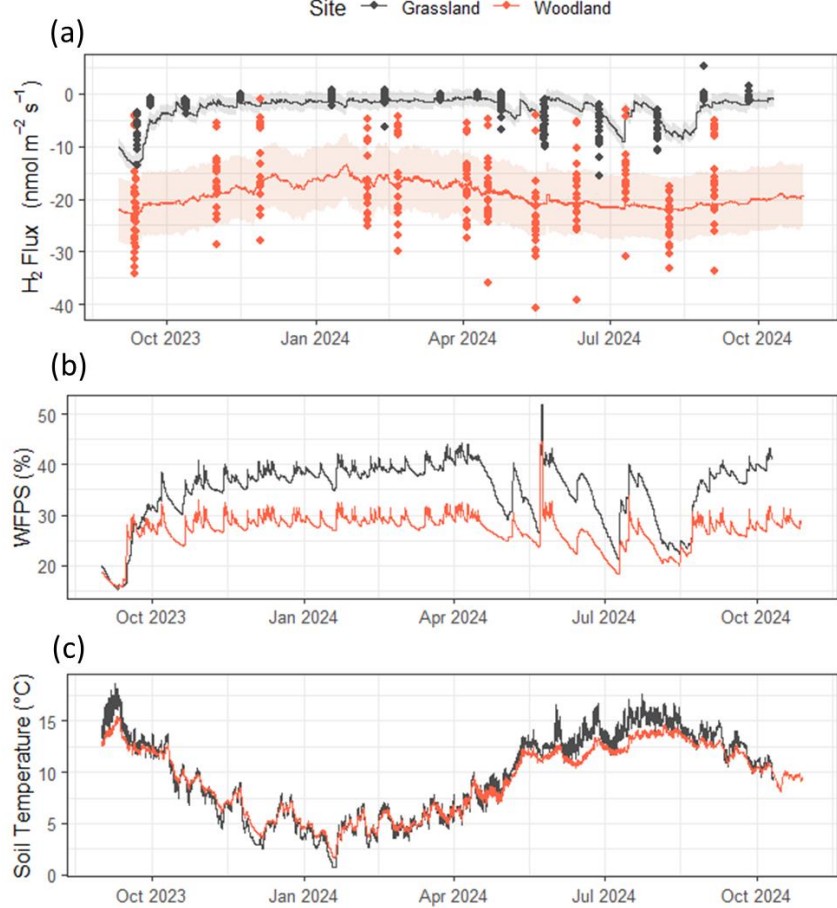


**Figure 4.** (a) $H_2$ flux measurements and model predictions for both field sites using a multiple regression fit
with soil moisture ($x$) and soil temperature ($z$) ($y = a_1x^2 + a_2x + b_1z^2 + b_2z + c$). (b) Continuous water filled pore
space (WFPS) at measurements made at 10 cm depth (average of 60 mins). (c) Continuous soil temperature
at measurements made at 10 cm depth (average of 60 mins).



## 4. Discussion

### *4.1. Quantification of H₂ flux*

Fluxes of $H_2$ measured in this study range from -40.7 to 5.3 nmol m$^{-2}$ s$^{-1}$ with mean fluxes of -2.6 ± 0.4 and -18.7 ± 1.0 nmol m$^{-2}$ s$^{-1}$ for the grassland and woodland sites, respectively. Using regression to model (gap-fill) flux data, we estimate annual $H_2$ uptake of 3.1 ± 0.1 and 12.0 ± 0.4kg $H_2$ ha$^{-1}$ yr$^{-1}$ for the grassland and woodland sites, respectively, which increases the expected mean uptake at the grassland site to 4.3 ± 0.2 nmol m$^{-2}$ s$^{-1}$ while the expected mean uptake at the woodland site remains near 18 nmol m$^{-2}$ s$^{-1}$. Predicted uptake is higher at the grassland site due to the expectation in the model that uptake will increase during periods of drier soils that were not measured directly. Predicted uptake estimated by the model and the extrapolation of the mean flux are not significantly different at the woodland site due to the lack of correlation with soil drivers in the model. However, the model does predict that uptake will slow down during the coldest months when fewer measurements were made at the site.

Mean measured uptake of $H_2$ at the grassland site is at the lower end of uptake reported in other studies that directly measured $H_2$ flux from soils, which range from -1.5 to >20 nmol m$^{-2}$ s$^{-1}$ (Table 3). The mean soil uptake of $H_2$ at the woodland site is at the higher end in terms of uptake magnitude, close in magnitude to high deposition velocities reported for peatlands in Simmonds et al., (2011). While uptake at this site seems high, we are confident that the flux measurements are accurate based on the consistency of flux observations and the quality controls put in place. Concentrations of $H_2$ in the chambers consistently fell exponentially, reaching near zero within 5 minutes (often within 3 mins) of enclosure. At the time of chamber closure ($t_o$), a volume of 0.025 m$^3$ of ambient air at the woodland site contains approximately 400-500 nmol of $H_2$ gas. To reach zero within 5 mins would require fluxes approximately 10-12 nmol m$^{-2}$ s$^{-1}$ in magnitude. While dealing with the exponential non-linearity of the rate of change of the concentration ($dC/dt$) does introduce an element of uncertainty in the flux calculations, we are confident the method used in this study (HMR fitting) accurately captures the flux at $t_o$ and thus a realistic magnitude of soil $H_2$ uptake.

Only two of the measured $H_2$ fluxes were both positive and larger than the analytical noise of the measurement method. However, these measurements from separate chambers on separate dates (from the grassland site) both showed 7 consecutive concentration measurements, all clearly increasing with time, highlighting that it is possible for $H_2$ emissions to occur in soils, even where uptake is the predominant direction of flux. It has been observed that legumes produce $H_2$ during the nitrogen fixation process (e.g. Schubert and Evans 1976; Flynn et al., 2014); however, no legume plants were present in any of the chamber locations during the study. The source of these $H_2$ emissions remains unknown and at no point did either of the field sites become a source of $H_2$, but our observations do highlight that there remain unknown microbial processes at the sub-field scale.



**Table 4.** A summary of $H_2$ fluxes and deposition velocity (Vd) measurements reported in literature, compared
with measured and modelled values in this study. Mean values and reported uncertainties. Where only flux
or Vd is reported, missing values are estimated using an ambient $H_2$ concentration of 500 ppb.

| Study | Soil Type | Country | Mean $H_2$ Flux (nmol $m^{-2}$ $s^{-1}$) | Mean Vd (cm $s^{-1}$) |
|---|---|---|---|---|
| This study (measured) | Grass (Grazing) | UK (SCO) | -2.6 ± 0.4 | 0.012 ± 0.002 |
| This study (gap-filled annual average) | Grass (Grazing) | UK (SCO) | -4.3 ± 0.2 | |
| This study (measured) | Woodland | UK (SCO) | -18.2 ± 1.0 | 0.088 ± 0.005 |
| This study (gap-filled annual average) | Woodland | UK (SCO) | -18.7 ± 0.6 | |
| Smith-Downey et al. (2008) | Forest | USA (CA) | -7.9 ± 4.2 | 0.063 ± 0.029 |
| | Desert | USA (CA) | -7.6 ± 5.3 | 0.051 ± 0.036 |
| | Marsh | USA (CA) | -7.5 ± 3.4 | 0.035 ± 0.013 |
| Lallo et al. (2009) | Urban park | FIN (Hesa) | -10.0 ± 2.5 | 0.020 ± 0.005 |
| | Urban park | FIN (Hesa) | -19.0 ± 3.5 | 0.038 ± 0.007 |
| Hammer and Levin (2009) | Urban/Agriculture | GER (BW) | -6.4 ± 1.6 | 0.03 ± 0.007 |
| Simmonds et al. (2011) | Peatland | IRE (GAL) | 26.5 (9.0 – 64.5) | 0.053 (0.018 – 0.129) |
| Meredith et al. (2017) | Woodland | USA (MA) | -3.2 ± 1.6 | 0.003 to 0.043 |
| Baril et al. (2022) | Arable | CAN (QC) | -5.9 ± 4.3 | 0.012 ± 0.009 |
| Buzzard et al. (2022) | Desert (Monsoon) | USA (AZ) | -1.5 to -3.5 | 0.03 to 0.007 |
| Nagai et al. (2024) | Arable | JAP (JP02) | -5 to -10 | 0.01 to 0.02 |


### 4.2. Drivers of $H_2$ flux

This study provides evidence of large variability in $H_2$ flux behaviour across two different soil types and the
importance of environmental factors such as soil temperature and moisture content. At the grassland site,
soil moisture (WFPS) dominated the $H_2$ flux behaviour in the soils. The relationship between $H_2$ uptake and
soil moisture was statistically significant (p <0.001) and explained 60% of the variance observed in the
grassland $H_2$ fluxes observed. While $H_2$ flux does appear to correlate with soil temperature at the grassland
site when compared directly, this is almost entirely due to the strong correlation between soil moisture and
soil temperature ($R^2$ = 0.68). Multiple regression finds soil temperature to be an insignificant variable once
the effect of soil moisture is accounted for at the grassland site. Spatial variability in $H_2$ fluxes at the woodland
site were an order of magnitude higher than those at the grassland site. This spatial variability could not be
explained by soil moisture, temperature or the total carbon content of the soil. While there do appear to be
weak relationships between the flux data and soil moisture and soil temperature, neither is found to be
statistically significant (maximum p-value of 0.15 for soil temperature).
Meteorological conditions were almost identical at the local scale (sites are less than 3 km apart) and soil at
both sites was of a similar pH and had similar total carbon and nitrogen contents. A small difference in



ambient $H_2$ concentrations was observed between the sites which may be caused by the large soil uptake and
poorer circulation of air at the woodland site, resulting in lower near surface $H_2$ concentrations. The reason
for the large difference in flux of $H_2$ measured between the two sites is not entirely clear from the measured
data, but it is likely that the physical properties of the soils played a role. While rooting systems and carbon
structure within the surface layers of the soils will be different at the sites, one large and obvious disparity is
the clay content of the soils which is approximately 55% and 25% at the grassland and woodland sites,
respectively. The higher density clay soil of the grassland site results in the soil becoming highly anaerobic
when moisture levels increase, as can be seen in the switching from $CH_4$ uptake to $CH_4$ emission when WFPS
exceeded 40%. At the woodland site, a thin layer of organic materials (forest litter) lies on top of a sandy, well-
drained soil, which may provide ideal conditions for $H_2$ uptake. Uptake of $CH_4$ is generally greater than at the
grassland site, and WFPS remains lower throughout the year, showing that drainage is significantly faster at
the site and suggests that the soils are more aerobic than at the grassland site (e.g. better penetration of $H_2$
to active regions within the soil). While the differences in soil texture may partly explain the large magnitude
of difference in $H_2$ uptake between the sites, it does not account for the large spatial variability of $H_2$ flux at
the woodland site. While the flux at the grassland site is largely dependent on physical factors at the field
scale such as the moisture content (aeration) of the soil, the woodland site showed large variations between
plots. This variation may be due to microbial factors that are highly spatial in a forest floor, such as available
nutrients (labile carbon from rotting plant litter), canopy shading and varying microbial densities.

### 4.3. Considerations for future research

Chamber flux methods are commonplace in the field of GHG flux measurements, but there are several
important factors that need to be considered when carrying out $H_2$ flux measurements in the field. One of
the most important - when using gas chromatography analysis - is the lifetime of samples stored in vials.
While it is possible to keep GHG samples in these vials for weeks or even months without significant storage
loss, $H_2$ concentrations were found to change relatively quickly, and should be analysed as soon as is possible
(within 24 h of measurement). This severely limits the reach of a particular field experiment to within travel
distances of a working $H_2$ gas chromatography instrument (e.g. not suitable for international shipment of
samples). Almost all published $H_2$ flux measurements to date are within the temperate region of the northern
hemisphere (USA and Europe), which limits the available data for models to predict soil/atmosphere
interactions at the global scale. Building $H_2$ flux datasets at a global level would require either investment in
localised infrastructure that allows for samples to be analysed in-country, or for the deployment of temporary
roving measurement methodology which travels between sites. We emphasise that unless particular care and



attention is applied to the transportation of gas samples (e.g. tests and quality control checks), the $H_2$ flux
cannot be analysed over a large distance due to leakage of samples.
Field measurements of $H_2$ are beneficial due to realistic environmental conditions. However, the manual
aspects of chamber sampling create logistical issues (extensive fieldwork) and the overlap of many
environmental and soil variables can make it difficult to identify the driving forces behind $H_2$ flux (e.g. the soil
moisture/temperature comparison). With this setup, the GC-PDHID is limited to one gas sample every 4
minutes, thus auto-chambers (chambers that open/close and measure gas samples automatically) are limited
in capability. New faster instruments able to measure $H_2$ gas via infra-red spectroscopy (by converting $H_2$ to
$H_2O$) are becoming more commercially available (see aerodyne.com/laser-analyzers), but there are no studies
using these analysers to date. Previously gas chromatography instrumentation has been used to measure $H_2$
flux via the aerodynamic gradient method (Meredith et al., 2017), which allows half hourly fluxes to be
measured at the field scale. While micrometeorological methods such as the aerodynamic gradient method
allow for a greater temporal and spatial coverage of soil fluxes, they also require certain field conditions, such
as flat open terrain and large (mains) power supply. In the case of the woodland site in this study,
micrometeorological methods are not feasible. With current available $H_2$ measurement methods, care must
be given when planning measurement activities to ensure efficiency in data collection.
Lab-based incubation studies of $H_2$ flux in literature are similar in number to those measured in the field.
Incubation studies allow for better control of soil conditions such as moisture, temperature and nutrient
content, environmental conditions (air temperature) and also for consistency in microbial populations (via
replicates of well mixed/homogenised soils). For example, in this study, it was difficult to determine the
impact of soil temperature due to the correlation with soil moisture. Due to the climate in the region, there
were no periods when the soils were cold and also dry, preventing observations of different extremes of the
driving forces behind $H_2$ flux. Incubation studies would be able to provide more information on these drivers
which may help modelling efforts; however, field measurements are still required to validate flux models as
incubation studies inevitably come with the caveat that flux measurements are not representative of true soil
conditions due to soil cores being repacked and creating therefore artificial conditions.

## 5. Conclusions

This study reports that the soil sink (uptake) of $H_2$ for a grassland and a forest site in close proximity is -3.1 ±
0.1 and -12.0 ± 0.4 kg $H_2$ ha$^{-1}$ yr$^{-1}$, respectively (with mean Vds of 0.012 ± 0.002 and 0.088 ± 0.005 cm s$^{-1}$ for
grassland and forest soils, respectively).  Soil moisture was found to be the primary driver of $H_2$ uptake at the
grassland site, where the high clay content of the soil resulted in anaerobic conditions (near zero $H_2$ flux)
during wet periods of the year. Uptake of $H_2$ at the forest site was highly variable and did not correlate well



with any localised soil properties. Both sites were exposed to similar meteorological conditions (3 km apart)
and had similar basic soil properties (such as pH and carbon content), thus we conclude that the large
difference in uptake between the soils was dependent on soil aeration. It is likely that the high clay content
of the grassland site (55%) resulted in a lack of aeration when soils were wet, while the well-drained forest
site (25% clay) was not restricted by exchange of $H_2$ between the atmosphere and the soil, showing instead a
large variability in $H_2$ flux that could be related to heterogeneous factors that control microbial activity (e.g.
labile carbon and microbial densities). In order to account for the large magnitude of site-scale differences
like those observed in this study, further field sites should be studied over a range of soil and land cover types
and management activities to improve global models of the soil $H_2$ sink. In addition, laboratory incubations
are needed to measure $H_2$ fluxes under controlled environmental conditions to refine the main driving
parameters of $H_2$ fluxes further.

## 6. Acknowledgements
Funding for this study has been provided by the UKRI Natural Environment Research Council (NERC) under
Grant Ref: NE/X013456/1 (Topic B: The Enigma of the Soil Hydrogen Sink Variability [ELGAR]). The work has
also been supported by the UK Research and Innovation (UKRI) Global Challenges Research Fund (GCRF) as
part of the South Asia Nitrogen Hub SANH project (NE/S009019/1).

## 7. Competing interests

The authors declare that they have no conflict of interest.

## 8. Data availability

Data currently undergoing preparation for submission to the Environmental Information Data Centre (EIDC).
https://eidc.ac.uk/



## 9. Author contributions

N. Cowan was the primary author of the manuscript and carried out all data analysis presented. The field team that developed measurement methodology protocols, carried out measurements, maintained field instrumentation and performed lab analysis consisted of T. Roberts, M. Hanlon, A. Bezanger, G. Toteva, A. Tweedie, K. Yeung and A. Deshpande. The project management and significant contributors to the manuscript text consisted of P. Levy, U. Skiba, E. Nemitz and J. Drewer. All coauthors contributed to the writing of the manuscript before submission.

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
