# Peer review of "Quantifying the soil sink of atmospheric Hydrogen: a full year of field measurements from grassland and forest soils in the UK"

_EGUsphere, 2024_

## Author Response (AR1)

Quantifying the soil sink of atmospheric Hydrogen: a full year of field measurements from grassland and forest soils in the UK - EGUSPHERE-2024-3654

In response to the comments of the associate editor, please find our replies and corrections below.

*Both reviewers commented on lines 55-56 ("The soil H2 sink is caused by microbial activity, both under aerobic and anaerobic conditions"). They point out that the atmospheric H2 soil sink is almost entirely aerobic since it occurs at the atmosphere-biosphere interface. The reworded text ("The soil sink of atmospheric H2 is caused by microbial activity, both under aerobic and anaerobic conditions") does not address the comments. The cited reference indicates that microbial consumption of H2 can occur under anaerobic as well as aerobic conditions, although the current soil sink of atmospheric H2 remains almost entirely under aerobic conditions.*

Text changed to:

Whilst both removal mechanisms are highly uncertain (especially the soil sink), the fate and impact of increased $H_2$ emissions depends largely on the soil sink strength (Ehhalt & Rohrer, 2009). The microbial uptake of $H_2$ can occur both under aerobic and anaerobic conditions, but the global atmospheric $H_2$ sink is dominated by processes that occur in aerobic soils at the atmosphere-biosphere interface (soil surface) where atmospheric $H_2$ availability is not as limited (Piché-Choquette & Constant, 2019).

*Lines 68-71 (Reviewer 1): The reworded text needs a semicolon instead of a comma.*

Corrected.

*The reference in line 73 referred to the spatial variation in microbial diversity (Reviewer 1). This reference (Baril et al.) is specifically about H2 flux variability in a winter cover crop study. Thus, the generalized statement (including "H2 measurement data are limited") should be revised to reflect more accurately the conclusions of the cited reference.*

Text reworded:

The biological sink of atmospheric $H_2$ has been suggested to be more sensitive to spatial variations of drivers (specifically microbial diversity) compared to the fluxes of other gases with high variability such as nitrous oxide ($N_2O$) (e.g. Baril et al., 2022); however, studies reporting the spatial variability of $H_2$ fluxes in soils are limited.

*Lines 255-260 (Reviewer 1) remain unclear, even after the minor revision. Please consider a more significant revision than what was provided.*

We hope this clarifies a bit better:

. No correlation between $H_2$ flux with measured total soil C or N in the top 10 cm was found at the woodland site ($R^2 < 0.01$ for each) (Figure S5). Variability in C and N in the replicated cores in the soils in the vicinity of each chamber (< 1 $m^2$ distance) was similar to the magnitude of

Quantifying the soil sink of atmospheric Hydrogen: a full year of field measurements from grassland and forest soils in the UK - EGUSPHERE-2024-3654

spatial variability observed at the entire plot scale. This suggested a relatively large variability in the soil C and N content at small scales which may obfuscate correlation between soils and fluxes at the individual chamber scale (destructive sampling could not be carried out on soil within the chambers without invalidating flux measurements).

*Lines 208-210 (Reviewer 2): While it does not make sense to have a negative deposition velocities, one should not exclude the reporting of positive fluxes. It appears from this study that there may be simultaneous production and consumption of H2, as indicated by the positive and negative fluxes. Thus, 'fluxes' should be referred to as 'net fluxes' throughout the manuscript.*

By definition a flux already represents the net direction of positive and negative motion. To clarify we have added the term "net flux" where we have carried out averaging to data where individual positive / negative fluxes have been combined.

*Lines 295-296 (Reviewer 1). The practice of using positive 'uptake' fluxes to mean the same thing as negative fluxes can be confusing, especially when it occurs in the same sentence.*

Flux changed to uptake and signs reversed.

*Lines 347-350 (Reviewer 1). How can both soils have 'similar particle density', when they have very different clay contents? Does this imply that the particle density of clay is similar to the particle density of sand and silt?*

Particle density is the density of the materials. If the sand/silt and clay are made from the same minerals, the particle density of the soil will be the same regardless of the composition. (Used to calculate bulk density, most studies use a predefined value of 2.65 g cm$^{-3}$ instead of measuring it.)

However, the clay/silt composition for the EB site is part of a correction that is applied now throughout the paper. The original sand/silt/clay ratios reported to us were wrong for the EB site. The silt/clay fraction was reversed, making the site high silt instead of high clay. This has changed the text in several areas of the manuscript but doesn't change the outcome of the study.

*Line 181-183 (AE comment): The spurious fits to equation 1 as shown by Examples 1 and 2 cannot be due to the 'low concentrations near the limit of the detection of the analyzer'. The concentrations shown are near-ambient. These may be due to being below the limit of flux detection of the analyzer, owing to instrumental precision limits. Or it may be because of simultaneous production and consumption obfuscating the gross uptake.*

Quantifying the soil sink of atmospheric Hydrogen: a full year of field measurements from grassland and forest soils in the UK - EGUSPHERE-2024-3654

Apologies, this was a poorly worded bit of text which confuses work we are doing on flux limits of detection with concentrations.

Text changed to:

Where soil flux is near the analytical uncertainty of the method (e.g. concentration change within the chamber is difficult to detect within the limitation ofthe instrument), a clear exponential decline was hard to discern from the measurement noise and could give rise to spurious fits to Equation 1. (Examples 1 and 2 in Figure 1 and Table 2).

The analytical uncertainties reported in the manuscript have been recalculated since the development of the equation which is now cited:

Cowan, N., Levy, P., Tigli, M., Toteva, G., Drewer, J., 2025. Characterisation of Analytical Uncertainty in Chamber Soil Flux Measurements. European J Soil Science. https://doi.org/10.1111/ejss.70104

**Additional corrections:**

Clay/Silt % of the EB site and resultant text changed throughout manuscript, including Table 1.

Figure 1 text changed from "analytical uncertainty" to "instrument noise"

Figure 1. Examples of concentration data collected during $H_2$ flux chamber sampling. Linear regression (grey) and HM model (brown) are used to determine dC/dt for each chamber measurement. Error bars represent the instrumental noise of $H_2$ measurements in GC analysis (15 ppb in this study). Comparisons of flux data presented in Table 2.

In response to the request for contour plots, this has proven difficult to show sensibly as the data does not cover a wide range of conditions at the sites, so extrapolation of missing data is extreme in some cases (Example below). Instead, we provide a scatter plot which has the same information, but is hopefully a bit easier to comprehend (now S4).

Quantifying the soil sink of atmospheric Hydrogen: a full year of field measurements from grassland and forest soils in the UK - EGUSPHERE-2024-3654

[Figure]

**Figure S4.** Scatterplots of WFPS and soil temperature at the Grassland and Woodland site.

Quantifying the soil sink of atmospheric Hydrogen: a full year of field measurements from grassland and forest soils in the UK - EGUSPHERE-2024-3654

[Figure]

Example contour plot of soil moisture/soil temperature data with flux. Note the missing data in the extremes of the grid produce unrealistic interpolations which limit the value of the projections. The high variability in flux in the limited range in which we have observations limits the value of a heat map, hence this data is now shown in S4.